# Multiple Human Papilloma Virus (HPV) Infections Are Associated with HSIL and Persistent HPV Infection Status in Korean Patients

**DOI:** 10.3390/v13071342

**Published:** 2021-07-12

**Authors:** Moonsik Kim, Nora Jee-Young Park, Ji Yun Jeong, Ji Young Park

**Affiliations:** Department of Pathology, Kyungpook National University Chilgok Hospital, Kyungpook National University School of Medicine, Daegu 41566, Korea; teiroa83@knuh.kr (M.K.); pathpjy@naver.com (N.J.-Y.P.); jjiyun@gmail.com (J.Y.J.)

**Keywords:** HPV, multiple infection, HPV genotype, cervical neoplasia

## Abstract

Infections with multiple human papilloma virus (HPV) types have been reported, but their role in cervical carcinogenesis has not been fully elucidated. In this study, 236 cases with multiple HPV infection were examined and compared to 180 cases with single HPV infection. HPV genotyping was performed with cervico-vaginal swab specimens using multiplex (real-time) polymerase chain reaction (PCR). In multiple HPV infection, the most prevalent HPV genotype was HPV 53, followed by HPV 16, 58, 52, and 68. HPV 33, 35, 39, 51, 52, 53, 58, and 68 were high-risk-HPV (HR-HPV) genotypes that were more frequently detected in multiple HPV infection compared to that in single HPV infection. The association between multiple HPV infection and high-grade SIL (HSIL) was significantly stronger compared to that of single HPV infection and HSIL (*p* = 0.002). Patients with multiple HPV infection displayed persistent and longer duration of the HPV infection compared to patients with single HPV infection. Multiple HPV infections have distinct clinicopathologic characteristics. Since it is associated with persistent HPV infection, HSIL, and different HR-HPV strains in contrast to single HPV infection, the presence of multiple HPV infection should be reported; close follow up is warranted.

## 1. Introduction

Human papillomavirus (HPV) infection is the major cause of cervical cancers and their precursor lesions [1,2]. Thus far, >200 HPV genotypes have been identified [3,4]. It is now evident that specific high-risk (HR) types of HPV cause the majority of squamous cell carcinomas (SQCCs) of the cervix as well as their precursors [5], although there are a few HPV-independent cervical SQCCs [6,7]. To date, >18 anogenital HPVs have been classified as oncogenic, these include HPV 16, 18, 26, 31, 33, 35, 39, 45, 51, 52, 53, 56, 58, 59, 66, 68, 69, 73, and 82 [8,9]. Although HPV 16, 18, 31, and 51 are common HR-HPV genotypes [9,10], the distribution pattern of HPV genotypes shows diverse regional variation [11,12,13,14]. For example, HPV 18, 52, and 58 are reported to be more prevalent in Asian populations [15].

Infection with multiple HPV types is reported in 20–45% of infected women [13,16,17], depending on the method of diagnosis [18,19]; however, the clinicopathologic significance remains debated, which can be attributed to the methodological diversities, lack of standardization of HPV tests, and indefinite selection from the test identities [20,21,22]. However, the results of some studies were significant; Brot et al. reported that multiple HR-HPV infection is associated with persistent low-grade squamous intraepithelial lesion (LSIL) and may help to identify patients at a higher risk of progression to high-grade squamous intraepithelial lesion (HSIL) and cervical SQCC [23]. In contrast, Salazar et al. suggested that multiple HPV infections may trigger intergenotypic competition and immune response, and thus do not contribute to the development of SIL [24].

HPV vaccines are a breakthrough in the prevention of major cervical cancers and greatly reduce cervical cancer rates. At present, three types of commercial vaccines are available: bivalent (HPV 16, 18), quadrivalent (HPV 16, 18, 6, 11), and nine-valent (HPV 16, 18, 6, 11, 31, 33, 45, 52, 58) vaccines [25,26]. It is essential to understand the nature of multiple HPV infection to establish prevention methods, including vaccination. In this study, we evaluated the clinicopathologic characteristics of multiple HPV infection, including infection prevalence, distribution of HPV genotype, and the association between cervical cancer and their precursor lesions compared to single HPV infection.

## 2. Materials and Methods

### 2.1. Study Population

Among 1967 cases wherein HPV genotyping tests were performed, we retrospectively found 236 consecutive cases (236/1967, 11.9%) with confirmed multiple HPV infection between 2017 and 2019 (Figure 1). For the control group, 180 cases of single HPV infection in the same period were randomly selected. Follow-up HPV genotyping tests were performed in each 6–12 month interval. All the patients underwent HPV genotyping tests because of health check-ups, abnormal cytology tests, or biopsy results. Clinicopathologic data of the patients including age, cytology and biopsy results, and duration of HPV infection were retrieved from their medical records.

### 2.2. Cytological Testing

Liquid-based Papanicolaou tests (SurePath; BD, Franklin Lakes, NJ, USA) were performed for cytological evaluation. Two experienced gynecologic pathologists (JYJ and NJP) at Kyungpook National University Chilgok Hospital interpreted the cytologic findings, as per the 2001 Bethesda System for cervicovaginal cytology reporting.

### 2.3. HPV Genotyping

HPV genotyping was performed with cervicovaginal swab specimens using the Anyplex II HPV28 assay kit (Seegene, Seoul, Korea). DNA extraction was carried out following the manufacturer’s instructions. Briefly, 5 μL of DNA was used in each of two 20 μL reactions with primer set A or B. In the assay, HPV-specific dual priming oligonucleotides were used for multiplex (real-time) polymerase chain reaction (PCR). A total of 28 HPV types were tested to simultaneously detect 18 HR types (HPV16, 18, 26, 31, 33, 35, 39, 45, 51, 52, 56, 58, 59, 66, 68, 69, 73, and 82) and 8 low-risk types (HPV 6, 11, 40, 42, 44, 53, 54, and 70). Persistent HPV infection was defined as the detection of any HPV genotype in two consecutive genotyping tests (first test and a follow-up test) [27]. The follow-up interval was 6–12 months for most patients (average follow-up interval: 10.2 months).

### 2.4. Pathologic Evaluation

From a total of 416 cases, comprising 236 multiple HPV infections and 180 single HPV infections detected by HPV genotyping tests, 264 cases with abnormal cervical screening tests (162 multiple and 102 single HPV infection cases) underwent colposcopy-guided cervical biopsy (Figure 1). Cervical cytology results corresponding to ASCUS or worse were regarded as abnormal. HPV 16/18 positivity with negative cervical cytology results was also considered abnormal according to the ASCCP consensus guidelines [28]. Cervical biopsies were gathered during the entire follow-up, including the first consultation. Pathologic diagnosis on the corresponding tissue specimens was independently reviewed by two gynecologic pathologists (JYJ and NJP) in a blinded manner. Cases with discrepant results were repeatedly reviewed until a consensus was reached.

### 2.5. Statistical Analysis

Relationships between clinicopathologic parameters were evaluated using the chi-square test for categorical parameters and Fisher’s exact test for parameters with an expected frequency of <5. We used the Kaplan–Meier method to evaluate the duration of HPV infection. Statistical differences in the duration of the infection were determined using the log-rank test. Differences were considered significant at *p* < 0.05. For multiple comparisons, an adjusted p-value was applied using the Bonferroni correction. All statistical analyses were conducted using IBM SPSS Statistics v.23 (IBM, Armonk, NY, USA).

## 3. Results

### 3.1. Overall Prevalence of Multiple and Single HPV Infection

The average age of the patients was 56.3 years in multiple HPV infection and 53.8 years in single HPV infection. HPV 53 (27.5%) was the most prevalent genotype in multiple HPV infection, followed by HPV 16 (21.8%), 58 (21.2%), 52 (20.8%), and 68 (20.8%). HPV 16 (15.6%) was the most prevalent genotype in single HPV infection, followed by HPV 56 (11.7%), 53 (8.3%), 52 (7.8%), and 68 (7.2%). Among HR-HPV, HPV 33 (*p* < 0.001), 35 (*p* < 0.001), 39 (*p* < 0.001), 51 (*p* < 0.001), 52 (*p* = 0.001), 53 (*p* < 0.001), 58 (*p* < 0.001), and 68 (*p* < 0.001) were more frequently detected in multiple HPV infection than in single HPV infection. Detailed information on HPV infection status is presented in Figure 2, Table 1, and Appendix A.

### 3.2. HPV Infection Status in Association with Histology

Next, we investigated the HPV infection status in association with the cervical lesion grades (Table 2). HSIL was associated significantly more with multiple HPV infection (76/236, 33.2%) compared to that of single HPV infection (33/180, 18.3%; *p* = 0.002). LSIL was also more frequently found in multiple HPV infection (37/236, 15.7%) compared to that in single HPV infection (18/180, 10.0%), although it was not statistically significant. There was no major difference in the proportion of SQCC or adenocarcinoma (ADC) between the single (30/180, 16.7%) and multiple HPV infection groups (29/236, 12.3%). SIL was found to be more frequently negative in single HPV infection group (94/180, 52.2%) compared to that in the multiple HPV infection group (91/236, 38.6%; *p* = 0.007). Most of the HSILs, SQCCs, or ADCs were associated with HR-HPV infection (Table 3); only six cases of single HPV infection (HPV 6, 42, 43, 54, 61, and 70) and one case of multiple HPV infection (HPV 6 and 42) progressed to HSIL, SQCC, or ADC without evidence of HR-HPV infection. In the multiple HPV infection group, HPV 16 was also the most prevalent HPV strain among HSIL (25/76, 32.9%) or SQCC (11/29, 37.9%) (Table 3). There was no specific co-infection pattern with HPV 16 and the other HPV strain. Although HPV 16 infection was most frequently associated with HSIL (32.9%) or SQCC (37.9%) in multiple HPV infection, other variable HR-HPV strains also contributed to HSIL or SQCC compared to the single HPV infection group. (Figure 3 and Appendix A). Patients with HSIL frequently had HPV 53 (27.6%), 58 (27.6%), 68 (25.0%), 66 (18.4%), 33 (18.4%), 35 (15.8%), and 39 (13.2%), and patients with SQCC had HPV 52 (20.7%), 18 (17.2%), 35 (17.2%), and 58 (13.8%) in addition to HPV 16 (Table 3).

### 3.3. Multiple HPV Types Were Associated with Persistent and Longer Duration of HPV Infection

We evaluated if infection with multiple HPV strains affects the persistence of HPV infection. Multiple HPV types were more significantly associated with persistent HPV infection than single HPV infection (180/193, 93.5% vs. 91/145, 62.8%, *p* < 0.001). Furthermore, we investigated if there was any correlation between multiple HPV infection and the duration of HPV infection. The median follow-up period was 26 months. The infection lasted significantly longer in patients with multiple HPV strains compared to patients infected by a single HPV strain (median period: 68 vs. 27 months, *p* < 0.001, Figure 4A). Longer durations of HPV infection in the multiple HPV genotype group were also identified in subgroups who had HSIL or SQCC (Figure 4B). Among the multiple HPV infection group, there was no association with the duration of HPV infection according to HPV genotypes (Appendix A) or combination patterns of HR-HPV or LR-HPV (Appendix A).

## 4. Discussion

In this study, we demonstrated that: (1) a diverse distribution of HPV genotypes occurs in multiple HPV infections in contrast to single HPV infections, (2) multiple HPV infections are more associated with HSIL and the duration of their infections is longer than that of single HPV infections, and (3) various HR-HPV strains contribute to HSIL or SQCC in addition to HPV 16.

The role of HPV infection in cervical carcinogenesis has been well-established [29]. Although the majority of HPV infections are transient, some HPV infections, especially HR-HPV and longer duration infection types, can progress to HSIL or SQCC [30]. Various risk factors can affect the progression to cervical carcinogenesis, including HPV genotypes, patient age, multiparity, socioeconomic status, and personal life style [31]. However, the role of multiple HPV infection in cervical carcinogenesis is still controversial and requires elucidation.

The prevalence and distribution of multiple and single HPV infections vary widely worldwide and are affected by diverse factors, including age, socioeconomic status, immune status, and vaccination status [15,32,33]. In this study, in decreasing order, HPV 53 (27.5%), 16 (21.8%), 58 (21.2%), 52 (20.8%), and 68 (20.8%) were the prevalent genotypes in multiple HPV infection among HR-HPV patients. HPV 53, 58, 52, 68, and other HR-HPV genotypes, which were frequently detected in the current study, were also commonly reported in previous Korean population-based HPV studies [14,34,35]. In addition, HPV 33 (*p* < 0.001), 35 (*p* < 0.001), 39 (*p* < 0.001), 51 (*p* < 0.001), 52 (*p* = 0.001), 53 (*p* < 0.001), 58 (*p* < 0.001), 59 (*p* = 0.009), 66 (*p* = 0.003), and 68 (*p* < 0.001) were associated more frequently with multiple than single HPV infection. As most of these genotypes were identified in a similar range of 10–20% without specific patterns of co-infection or phylogenetically related groups, it is difficult to determine whether this is the result of the synergistic or competitive effects of different HPV genotypes or the mere sum of incidental cumulative HPV co-infection.

In our study, HSIL (76/236, 33.2%) was significantly more frequent in multiple HPV infection than in single HPV infection (33/180, 18.3%, *p* = 0.002). LSIL was also more frequently detected in multiple HPV infection (37/236, 15.7%), although statistically not significant. A study by Schimitt et al. also reported that multiple HPV types were prevalent in LSIL and HSIL [36]. It is generally accepted that progression to HSIL and SQCC correlates with monoclonal expansion of HPV-infected host cells [29]; however, some previous studies have suggested that multiple HPV infections confer synergistic impact on the developing SIL [11,12,37], which supports the impact of multiple HPV infections on SIL in this study. Possibly, different HPV genotypes infect different cells in distinct lesions in the same patient; subsequent studies are warranted to validate this hypothesis.

Although this study shows that HPV16 is the major cause of HSIL and SQCC, patients with multiple HPV infection also exhibited comparable amounts of other HR-HPV genotypes, including HPV 33, 35, 39, 53, 58, 66, and 68. Previous studies have demonstrated that the diverse distribution of prevalent genotypes and co-infection patterns of HR-HPVs in HSIL and SQCC depends on demographic and socioeconomic factors [38,39,40]. Bolaños et al. reported that HPV 51 and 52 co-infections were common in the Mexican population in addition to HPV 16 [38]. Trottier et al. indicated that in addition to HPV 16, co-infections involving HPV 58 appeared to increase cervical oncogenic risk in Brazilian populations [22]. It is possible that synergistic interactions among HR-HPVs other than HPV 16 genotype may have influenced cervical carcinogenesis in our cohort group. Differences regarding immunity status, vaccination status, demographics, and other possible risk factors in Korean patients, who constitute most of the patients in our study, may have affected the distribution of HPV genotypes in multiple HPV infection. Although few HSIL and SQCC had only LR-HPVs without evidence of HR-HPV infections in this study, this is probably due to the insufficiency of previous medical records.

Another noteworthy finding of this study is that patients with multiple HPV genotypes had persistent HPV infection during follow-up periods compared to patients with single HPV infection. Some previous studies have pointed to the possible association between multiple HPV genotypes and persistence of infection [11,20,23,41]. A possible mechanism is that synergistic interactions between multiple HPVs help persistent HPV infection [25,38]. Changes in immune status caused by multiple HPV infections may affect the duration of HPV infection as well [42]. The increased viral load caused by HPV multiplicity is another possible reason for the persistence HPV infection [11].

This study has some limitations. First, the frequency of multiple HPV infections and the distribution pattern of HPV genotypes being influenced by the detection method. AnyplexTM II HPV28 assay is a real-time PCR-based HPV detection method, which is more sensitive compared to the HPV DNA chip assay or the HPV hybrid capture DNA test [43]. Therefore, the prevalence of multiple HPV infections and commonly infected HR-HPV types can be estimated differently on the other detection platforms. In addition, this study was conducted in a single institution with a limited number of cases; there may be different HPV distribution patterns among different regions or countries. Thus, a larger cohort study is warranted to validate the result of this study. Because of the retrospective nature of this study, we were not able to clarify the exact mechanism of multiple HPV infections on the progression of SIL and persistent HPV infection. Thus, further in vitro and in vivo studies are warranted to strengthen the effects of multiple HPV infections suggested in this study.

In conclusion, we demonstrated distinct clinicopathologic characteristics of multiple HPV infection in comparison to single HPV infection regarding the progression of the infection to SIL and the duration of the infection. Multiple HPV infections were also enriched with diverse HPV genotypes in contrast to single HPV infections. As understanding the distribution pattern and infection mechanism of multiple HPV genotypes is critical for establishing the prevention plans and vaccination programs of HPV, subsequent studies are warranted to further validate the results of this study.

## Figures and Tables

**Figure 1 viruses-13-01342-f001:**
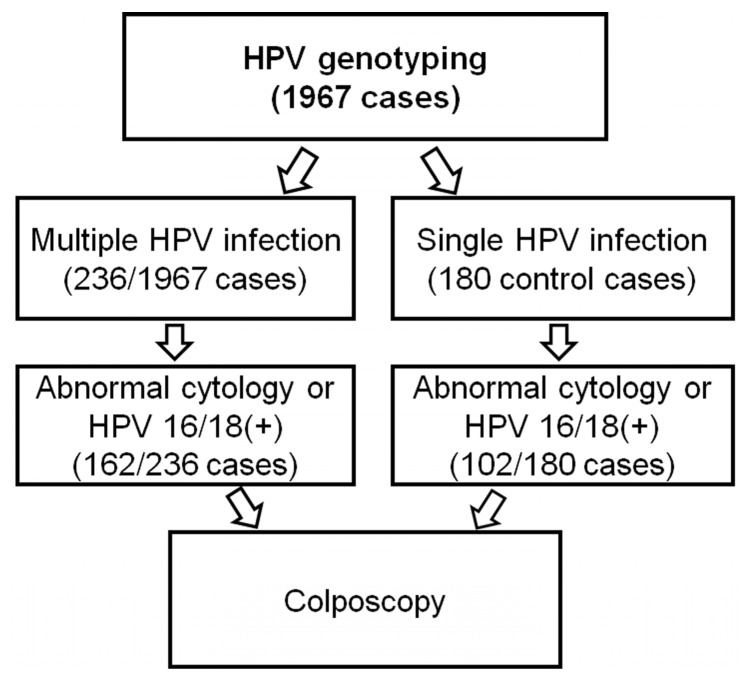
Selection of study cases.

**Figure 2 viruses-13-01342-f002:**
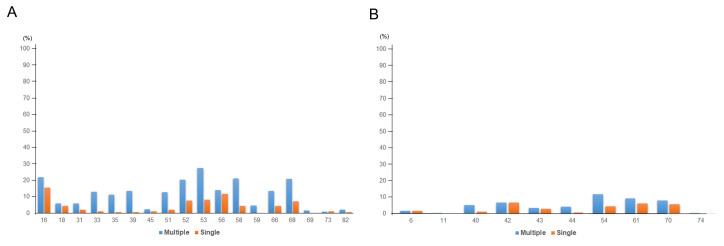
HPV genotype prevalence in single and multiple HPV infection (**A**) among HR-HPV and (**B**) LR-HPV.

**Figure 3 viruses-13-01342-f003:**
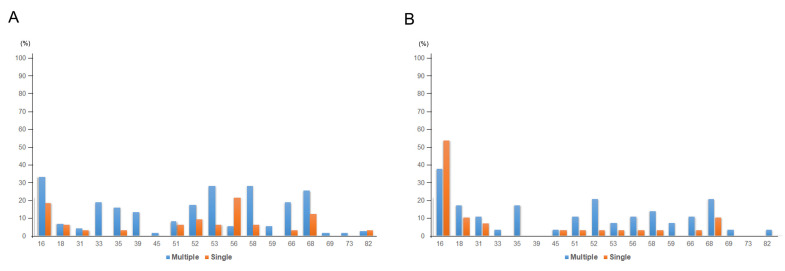
HPV genotype prevalence in (**A**) HSIL and (**B**) SQCC (single and multiple infection).

**Figure 4 viruses-13-01342-f004:**
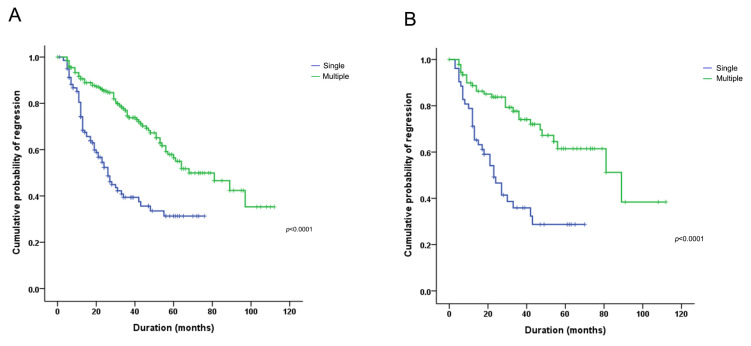
Kaplan–Meier curve of HPV infection during the follow-up period (in single and multiple HPV infections) (**A**) in all cases and (**B**) among those who had HSIL and SQCC. Regression was defined as the absence of HPV detection during any follow-up period.

**Table 1 viruses-13-01342-t001:** Distribution of HPV genotypes (single and multiple infection).

	Number	HPV Infection Status		*p*-Value
**Total**	416	Multiple (236)	Single (180)	
**Age (mean)**		56.3	53.8	0.065
**HR HPV**				
16		52 (22.0%)	28 (15.6%)	0.125
18		14 (5.9%)	8 (4.4%)	0.652
26		0 (0.0%)	1 (0.6%)	0.892
31		14 (5.9%)	4 (2.2%)	0.11
33		31 (13.1%)	2 (1.1%)	<0.001 ^a^
35		27 (11.4%)	1 (0.6%)	<0.001 ^a^
39		32 (13.6%)	1 (0.6%)	<0.001 ^a^
45		6 (2.5%)	2 (1.1%)	0.488
51		30 (12.7%)	4 (2.2%)	<0.001 ^a^
52		48 (20.3%)	14 (7.8%)	0.001 ^a^
53		65 (27.5%)	15 (8.3%)	<0.001 ^a^
56		33 (14.0%)	21 (11.7%)	0.583
58		50 (21.2%)	8 (4.4%)	<0.001 ^a^
59		11 (4.7%)	0 (0.0%)	0.009
66		32 (13.6%)	8 (4.4%)	0.003
68		49 (20.8%)	13 (7.2%)	<0.001 ^a^
69		4 (1.7%)	0 (0.0%)	0.212
73		2 (0.8%)	2 (1.1%)	1
82		5 (2.1%)	1 (0.6%)	0.363
**LR HPV**				
6		4 (1.7%)	3 (1.7%)	1
11		1 (0.4%)	0 (0.0%)	1
30		0 (0.0%)	0 (0.0%)	N/A
32		0 (0.0%)	0 (0.0%)	N/A
34		0 (0.0%)	0 (0.0%)	N/A
40		12 (5.1%)	2 (1.1%)	0.051
42		16 (6.8%)	12 (6.7%)	1
43		8 (3.4%)	5 (2.8%)	0.943
44		10 (4.2%)	1 (0.6%)	0.044
54		28 (11.9%)	8 (4.4%)	0.013
55		0 (0.0%)	0 (0.0%)	N/A
61		22 (9.3%)	11 (6.1%)	0.309
62		0 (0.0%)	0 (0.0%)	N/A
67		0 (0.0%)	0 (0.0%)	N/A
70		19 (8.1%)	10 (5.6%)	0.426
74		0 (0.0%)	0 (0.0%)	N/A
81		1 (0.4%)	0 (0.0%)	1
83		0 (0.0%)	0 (0.0%)	N/A
84		0 (0.0%)	0 (0.0%)	N/A
87		0 (0.0%)	0 (0.0%)	N/A
90		0 (0.0%)	0 (0.0%)	N/A

^a^ Significant *p*-value after Bonferroni correction (corrected *p*-value: 0.05/40 = 0.0125; all 40 HPV types were included) was found between single and multiple HPV infection groups.

**Table 2 viruses-13-01342-t002:** Correlation between histologic diagnosis and HPV infection status (single and multiple).

	No.	HPV Infection Status		*p*-Value
**Total**	416	Multiple (236)	Single (180)	
**Age (mean)**		56.3	53.8	
**Histology**				
Normal	185	91 (38.6%)	94 (52.2%)	0.007
LSIL	55	37 (15.7%)	18 (10.0%)	0.122
HSIL	110	76 (32.2%)	33 (18.3%)	0.002
Squamous cell carcinoma	63	29 (12.3%)	30 (16.7%)	0.26
Adenocarcinoma	8	3 (1.2%)	5 (2.8%)	0.454

**Table 3 viruses-13-01342-t003:** Association between histologic diagnosis and HPV genotypes (single and multiple HPV infection).

	Multiple HPV Infection					Single HPV Infection				
	Normal	LSIL	HSIL	SQCC	ADC	Normal	LSIL	HSIL	SQCC	ADC
	(*n* = 91)	(*n* = 37)	(*n* = 76)	(*n* = 29)	(*n* = 3)	(*n* = 94)	(*n* = 18)	(*n* = 33)	(*n* = 30)	(*n* = 5)
**HR HPV**										
16	13 (14.3%)	3 (8.1%)	25 (32.9%)	11 (37.9%)	0 (0.0%)	4 (4.3%)	0 (0.0%)	6 (18.2%)	16 (53.3%)	2 (40.0%)
18	4 (4.4%)	0 (0.0%)	5 (6.6%)	5 (17.2%)	0 (0.0%)	2 (2.1%)	0 (0.0%)	2 (6.1%)	3 (10.0%)	1 (20.0%)
31	5 (5.5%)	3 (8.1%)	3 (3.9%)	3 (10.3%)	0 (0.0%)	1 (1.1%)	0 (0.0%)	1 (3.0%)	2 (6.7%)	0 (0.0%)
33	10 (11.0%)	5 (13.5%)	14 (18.4%)	1 (3.4%)	1 (33.3%)	1 (1.1%)	1 (5.6%)	0 (0.0%)	0 (0.0%)	0 (0.0%)
35	7 (7.7%)	3 (8.1%)	12 (15.8%)	5 (17.2%)	0 (0.0%)	0 (0.0%)	0 (0.0%)	1 (3.0%)	0 (0.0%)	0 (0.0%)
39	14 (15.4%)	6 (16.2%)	10 (13.2%)	0 (0.0%)	2 (66.7%)	1 (1.1%)	0 (0.0%)	0 (0.0%)	0 (0.0%)	0 (0.0%)
45	2 (2.2%)	2 (5.4%)	1 (1.3%)	1 (3.4%)	0 (0.0%)	1 (1.1%)	0 (0.0%)	0 (0.0%)	1 (3.3%)	0 (0.0%)
51	15 (16.5%)	6 (16.2%)	6 (7.9%)	3 (10.3%)	0 (0.0%)	1 (1.1%)	0 (0.0%)	2 (6.1%)	1 (3.3%)	0 (0.0%)
52	16 (17.6%)	12 (32.4%)	13 (17.1%)	6 (20.7%)	1 (33.3%)	9 (9.6%)	1 (5.6%)	3 (9.1%)	1 (3.3%)	0 (0.0%)
53	27 (29.7%)	14 (37.8%)	21 (27.6%)	2 (6.9%)	1 (33.3%)	12 (12.8%)	0 (0.0%)	2 (6.1%)	1 (3.3%)	0 (0.0%)
56	19 (20.9%)	7 (18.9%)	4 (5.3%)	3 (10.3%)	0 (0.0%)	10 (10.6%)	3 (16.7%)	7 (21.2%)	1 (3.3%)	0 (0.0%)
58	21 (23.1%)	4 (10.8%)	21 (27.6%)	4 (13.8%)	0 (0.0%)	3 (3.2%)	2 (11.1%)	2 (6.1%)	1 (3.3%)	0 (0.0%)
59	3 (3.3%)	2 (5.4%)	4 (5.3%)	2 (6.9%)	0 (0.0%)	0 (0.0%)	0 (0.0%)	0 (0.0%)	0 (0.0%)	0 (0.0%)
66	9 (9.9%)	6 (16.2%)	14 (18.4%)	3 (10.3%)	0 (0.0%)	6 (6.4%)	0 (0.0%)	1 (3.0%)	1 (3.3%)	0 (0.0%)
68	19 (20.9%)	5 (13.5%)	19 (25.0%)	6 (20.7%)	0 (0.0%)	3 (3.2%)	2 (11.1%)	4 (12.1%)	3 (10.0%)	1 (20.0%)
69	1 (1.1%)	0 (0.0%)	1 (1.3%)	1 (3.4%)	0 (0.0%)	0 (0.0%)	0 (0.0%)	0 (0.0%)	0 (0.0%)	0 (0.0%)
73	1 (1.1%)	0 (0.0%)	1 (1.3%)	0 (0.0%)	0 (0.0%)	2 (2.1%)	0 (0.0%)	0 (0.0%)	0 (0.0%)	0 (0.0%)
82	1 (1.1%)	1 (2.7%)	2 (2.6%)	1 (3.4%)	0 (0.0%)	0 (0.0%)	0 (0.0%)	1 (3.0%)	0 (0.0%)	0 (0.0%)
**LR HPV**										
6	1 (1.1%)	1 (2.7%)	2 (2.6%)	0 (0.0%)	0 (0.0%)	1 (1.1%)	1 (5.6%)	0 (0.0%)	1 (3.3%)	0 (0.0%)
11	0 (0.0%)	1 (2.7%)	0 (0.0%)	0 (0.0%)	0 (0.0%)	0 (0.0%)	0 (0.0%)	0 (0.0%)	0 (0.0%)	0 (0.0%)
30	0 (0.0%)	0 (0.0%)	0 (0.0%)	0 (0.0%)	0 (0.0%)	0 (0.0%)	0 (0.0%)	0 (0.0%)	0 (0.0%)	0 (0.0%)
32	0 (0.0%)	0 (0.0%)	0 (0.0%)	0 (0.0%)	0 (0.0%)	0 (0.0%)	0 (0.0%)	0 (0.0%)	0 (0.0%)	0 (0.0%)
34	0 (0.0%)	0 (0.0%)	0 (0.0%)	0 (0.0%)	0 (0.0%)	0 (0.0%)	0 (0.0%)	0 (0.0%)	0 (0.0%)	0 (0.0%)
40	6 (6.6%)	2 (5.4%)	1 (1.3%)	3 (10.3%)	0 (0.0%)	1 (1.1%)	1 (5.6%)	0 (0.0%)	0 (0.0%)	0 (0.0%)
42	4 (4.4%)	2 (5.4%)	7 (9.2%)	2 (6.9%)	1 (33.3%)	11 (11.7%)	0 (0.0%)	1 (3.0%)	0 (0.0%)	0 (0.0%)
43	1 (1.1%)	2 (5.4%)	4 (5.3%)	1 (3.4%)	1 (33.3%)	1 (1.1%)	3 (16.7%)	0 (0.0%)	1 (3.3%)	0 (0.0%)
44	5 (5.5%)	0 (0.0%)	3 (3.9%)	2 (6.9%)	0 (0.0%)	1 (1.1%)	0 (0.0%)	0 (0.0%)	0 (0.0%)	0 (0.0%)
54	11 (12.1%)	2 (5.4%)	10 (13.2%)	4 (13.8%)	0 (0.0%)	5 (5.3%)	1 (5.6%)	0 (0.0%)	1 (3.3%)	1 (20.0%)
55	0 (0.0%)	0 (0.0%)	0 (0.0%)	0 (0.0%)	0 (0.0%)	0 (0.0%)	0 (0.0%)	0 (0.0%)	0 (0.0%)	0 (0.0%)
61	9 (9.9%)	2 (5.4%)	8 (10.5%)	2 (6.9%)	0 (0.0%)	9 (9.6%)	2 (11.1%)	0 (0.0%)	0 (0.0%)	0 (0.0%)
62	0 (0.0%)	0 (0.0%)	0 (0.0%)	0 (0.0%)	0 (0.0%)	0 (0.0%)	0 (0.0%)	0 (0.0%)	0 (0.0%)	0 (0.0%)
67	0 (0.0%)	0 (0.0%)	0 (0.0%)	0 (0.0%)	0 (0.0%)	0 (0.0%)	0 (0.0%)	0 (0.0%)	0 (0.0%)	0 (0.0%)
70	8 (8.8%)	4 (10.8%)	4 (5.3%)	3 (10.3%)	0 (0.0%)	8 (8.5%)	1 (5.6%)	1 (3.0%)	0 (0.0%)	0 (0.0%)
74	0 (0.0%)	0 (0.0%)	0 (0.0%)	0 (0.0%)	0 (0.0%)	0 (0.0%)	0 (0.0%)	0 (0.0%)	0 (0.0%)	0 (0.0%)
81	0 (0.0%)	0 (0.0%)	0 (0.0%)	1 (3.4%)	0 (0.0%)	0 (0.0%)	0 (0.0%)	0 (0.0%)	0 (0.0%)	0 (0.0%)
83	0 (0.0%)	0 (0.0%)	0 (0.0%)	0 (0.0%)	0 (0.0%)	0 (0.0%)	0 (0.0%)	0 (0.0%)	0 (0.0%)	0 (0.0%)
84	0 (0.0%)	0 (0.0%)	0 (0.0%)	0 (0.0%)	0 (0.0%)	0 (0.0%)	0 (0.0%)	0 (0.0%)	0 (0.0%)	0 (0.0%)
87	0 (0.0%)	0 (0.0%)	0 (0.0%)	0 (0.0%)	0 (0.0%)	0 (0.0%)	0 (0.0%)	0 (0.0%)	0 (0.0%)	0 (0.0%)
90	0 (0.0%)	0 (0.0%)	0 (0.0%)	0 (0.0%)	0 (0.0%)	0 (0.0%)	0 (0.0%)	0 (0.0%)	0 (0.0%)	0 (0.0%)

## Data Availability

All data generated and analyzed during this study are included in this article and its Appendix A.

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
