# Peer review of "Multiple Human Papilloma Virus (HPV) Infections Are Associated with HSIL and Persistent HPV Infection Status in Korean Patients"

_viruses, 2021, doi:10.3390/v13071342_

Round 1

Reviewer 1 Report

In the manuscript “Multiple human papilloma virus (HPV) infections are associated with HSIL and persistent HPV infection status in Korean patients”, by Kim et al. (viruses-1274349), the authors aim to evaluate the clinic-pathologic characteristics of HPV cervical infection in women respect to multiple or single HPV genotype infection. The authors describe the distribution of HPV genotypes, the association of multiple vs. single infection in respect to cervical cancer or cervical lesions, and in respect to persistence of infection. The manuscript describes the analysis of 236 consecutive cases characterized for presence of multiple HPV infection and 180 random selected cases of single HPV infection. HPV identification was based on Real-Time PCR from DNA isolated from cervical swab. Liquid-based cytology was used to identify women with abnormal cytology, which were referred to colposcopy and, possibly, biopsy for HSIL/LSIL and cancer diagnosis.

The manuscript has three major points that need to be clarified in respect to experimental design: (1) the authors need to define persistence of HPV infection in cases of single and multiple infection; (2) the authors need to clarify how persistence was evaluated, a more detailed description is required; (3) the authors need to better characterize progression and regression (item 3.3, page 8). The precise definition of these parameters is required to understand the manuscript.

Other points that need to be revised are:

- In respect to statistical analyzes,  the authors need to use a method to adjust the p value as multiple comparisons were carried out between cases of single and multiple infection in respect to the frequency of HPV genotypes.

- In statistical methodology (page 3, lines 95-104) the authors mention that an overall survival analysis was made, however there is no results in respect to this analysis.

-In respect to regression and persistence of infection (page 8, lines 154-163), it is not clear what is shown in Figure 4b. In the text is mentioned that this figure show a comparison of persistence and progression to HSIL and SQCC, however, it is not possible to observe these findings analyzing Figure 4b.

Author Response

Response to Reviewer #1

Thank you for giving us the opportunity to revise our manuscript. The reviewers’ comments were very helpful. Our responses to the comments are as follows.

Reviewer #1 (Comments to the Author):

Major comments

(1) the authors need to define persistence of HPV infection in cases of single and multiple infection; 

Thank you for this helpful comment. In this study, persistent HPV infection was defined as the detection of the HPV DNA in two consecutive genotyping tests, following the traditional definition (reference: Schiffman et al. J Natl Cancer Inst Monogr, 2003(31): p. 14-9.). We have added the definition of persistent HPV infection in revised manuscript (page 2, line 82-83)

(2) the authors need to clarify how persistence was evaluated, a more detailed description is required;

The persistence of HPV infection was evaluated using the definition discussed above. The follow-up intervals of most patients were between 6-12 months (average follow-up interval: 10.2 months). The differences in the persistence of HPV infection between single and multiple HPV tests were assessed using Chi-square test. The results were added in revised manuscript (page 8, line 157-158)

(3) the authors need to better characterize progression and regression (item 3.3, page 8).

Thank you for this comment. To avoid confusion, I deleted the word “progression” and revised the item 3.3 in the manuscript. In the revised item 3.3, the persistence of HPV infection was first compared between single and multiple HPV infectios. Then, the durations of these infections were evaluated using Kaplan-Meier method. Regression was defined as the absence of detected HPV in any of the follow-up periods (Page 8, line 170-171).

Other points

- In respect to statistical analyzes,  the authors need to use a method to adjust the p value as multiple comparisons were carried out between cases of single and multiple infection in respect to the frequency of HPV genotypes.

We adjusted the p-value using Bonferroni correction (corrected p-value: 0.05/4 = 0.0125; all 40 HPV genotypes were included) (page 3, line 102-103; Table 1).

- In statistical methodology (page 3, lines 95-104) the authors mention that an overall survival analysis was made, however there is no results in respect to this analysis.

In this study, Kaplan-Meier method was used to evaluate the differences in the durations of HPV infection, and not to evaluate the overall survival. Therefore, I revised the statistical analysis as follows (page 3, 100-102): Kaplan-Meier method was used to evaluate the duration of the HPV infection. Statistical differences in the durations of infection were determined using log-rank test.

-In respect to regression and persistence of infection (page 8, lines 154-163), it is not clear what is shown in Figure 4b. In the text is mentioned that this figure show a comparison of persistence and progression to HSIL and SQCC, however, it is not possible to observe these findings analyzing Figure 4b.

Figure 4B corresponds to the Kaplan-Meier analysis in the subgroup of single and multiple HPV infections ‘who had’ HSIL or SQCC on histologic diagnosis. The tendency for a longer duration of multiple HPV infection was also found in the subgroup who had HSIL or SQCC, which are of clinical concern. To avoid confusion, we used the expression “who had HSIL or SQCC” instead of who progressed into HSIL or SQCC (page 8, line 164).

Reviewer 2 Report

This paper describes the clinical and pathological characteristics of multiple HPV infections, including the infection prevalence, the distribution of HPV genotypes, and the association between cervical cancer and their precursor lesions compared with single HPV infection. Since this research brings a valuable data about HPV infection, the manuscript deserves to be published in “Viruses” journal. Also, the manuscript is clearly written, easy to read and understand. This reviewer has only a few minor suggestions:

  1. Page 2, line 48 – “and thus does not” should be changed to “and thus do not”
  2. Page 2, line 67 - “Among 1967 total cases” – please, delete “total”
  3. Page 2, line 67 – “genotyping tests was performed” should be changed to “genotyping tests were performed”
  4. Page 9, line 208 – please, add a space between “68.” And “Previous”
  5. Please, add a full stop after “et al” during the text (e.g. “Brot et al” should be changed to “Brot et al.”)
  6. Page 10, line 243 – “compared with” should be changed to “compared to” or “in comparison with”

Author Response

Reviewer #2 (Comments to the Author):

  1. Page 2, line 48 – “and thus does not” should be changed to “and thus do not”
  2. Page 2, line 67 - “Among 1967 total cases” – please, delete “total”
  3. Page 2, line 67 – “genotyping tests was performed” should be changed to “genotyping tests were performed”
  4. Page 9, line 208 – please, add a space between “68.” And “Previous”
  5. Please, add a full stop after “et al” during the text (e.g. “Brot et al” should be changed to “Brot et al.”)
  6. Page 10, line 243 – “compared with” should be changed to “compared to” or “in comparison with”

Thank you for the comments. We have revised the manuscript as your comment.

Round 2

Reviewer 1 Report

In the revised version of the manuscript “Multiple human papilloma virus (HPV) infections are associated with HSIL and persistent HPV infection status in Korean patients”, by Kim et al. (manuscript ID 1274349), the authors made changes in the text clarifying points of the previous version. The authors added a definition for persistence clarifying how persistence was evaluated. In the statistical analysis the authors added the correction for multiple tests (Bonferroni correction) and modified the text suppressing overall survival analysis. The authors also modified the item 3.3 (page 10) now entitled “Multiple HPV types were associated with persistent and longer duration of HPV” focusing this section on HPV persistence, consequently, the data presented in Figure 4b are now congruent with the text. The legend of figure 4 was also modified.

To make the manuscript more clear the authors need to clarify two points:

- When defining persistence as “the detection of HPV DNA in two consecutive genotyping tests (first test and a follow-up test)”, are the authors considering the presence of different or the same HPV genotypes in the first test and in the follow-up test?

- Were authors considering the data about the presence of LSIL, HSIL, and cancer those obtained only after the first HPV genotyping test?

Author Response

Response to Reviewer #1-Second round

Thank you for giving us the opportunity to revise our manuscript again. The reviewers’ comments were very helpful. Our responses to the comments are as follows.

Reviewer #1 (Comments to the Author):

- When defining persistence as “the detection of HPV DNA in two consecutive genotyping tests (first test and a follow-up test)”, are the authors considering the presence of different or the same HPV genotypes in the first test and in the follow-up test?

Thank you for the helpful comment. We defined persistent HPV infection as being positive for “any HPV genotype” at two consecutive genotyping tests. We have added it in the revised manuscript highlighted in emerald background (page 2, line 82).

- Were authors considering the data about the presence of LSIL, HSIL, and cancer those obtained only after the first HPV genotyping test?

- The data regarding the presence of LSIL, HSIL, and cancer was gathered through entire follow-up including the first consultation. By gathering the biopsies through entire follow-up, we thought that it can better reflect the actual time-dependent behavior of HPV virus. We also expected that it may lower the false-negativity on colposcopic biopsy result. We added it in the revised manuscript highlight in emerald background (page 3, line 91-92).